# Systematic Review on Efficacy, Effectiveness, and Safety of Pitavastatin in Dyslipidemia in Asia

**DOI:** 10.3390/healthcare13010059

**Published:** 2024-12-31

**Authors:** Nam Xuan Vo, Huong Lai Pham, Tan Trong Bui, Tien Thuy Bui

**Affiliations:** 1Faculty of Pharmacy, Ton Duc Thang University, Ho Chi Minh City 700000, Vietnam; h1900276@student.tdtu.edu.vn; 2Faculty of Medicine, University of Medicine and Pharmacy at Ho Chi Minh City, Ho Chi Minh City 700000, Vietnam; bttan.y22@ump.edu.vn; 3Faculty of Pharmacy, Le Van Thinh Hospital, Ho Chi Minh City 700000, Vietnam; bttien.ths.tcqld23@ump.edu.vn

**Keywords:** pitavastatin, dyslipidemia, hypercholesterolemia, systematic review, efficacy, effectiveness, safety

## Abstract

**Objectives:** Dyslipidemia, a significant risk factor for cardiovascular disease (CVD), is marked by abnormal lipid levels, such as the elevated lowering of low-density lipoprotein cholesterol (LDL-C). Statins are the first-line treatment for LDL-C reduction. Pitavastatin (PIT) has shown potential in lowering LDL-C and improving high-density lipoprotein cholesterol (HDL-C). This review assesses pitavastatin’s efficacy, effectiveness, and safety in dyslipidemia management in Asia. **Methods:** A systematic review was conducted using PubMed, Cochrane, and Embase databases up to November 2024, adhering to Preferred Reporting Items of Systematic Reviews and Meta-Analyses (PRISMA) guidelines. Seventeen studies (12 RCTs and 5 non-RCTs) were analyzed, focusing on LDL-C reduction, safety profiles, and adverse events. The quality of the studies was assessed using checklists to ensure the selection of the best studies and to limit bias. **Results:** Pitavastatin doses (1–4 mg) reduced LDL-C by 28–47%, comparable to atorvastatin, rosuvastatin, and simvastatin. The 2 mg dose matched atorvastatin’s 10 mg dose in efficacy for both short-term (35–42%) and long-term (28–36%) use. LDL-C target achievement rates were 75–95%. Adverse events, including mild myalgia and elevated liver enzymes, were rare, and discontinuation rates were low. **Conclusions:** Pitavastatin is an effective and safe alternative to traditional statins for dyslipidemia management in Asia. Further research on long-term outcomes and high-risk groups is warranted.

## 1. Introduction

Dyslipidemia is a condition marked by abnormalities in blood lipid levels, such as elevated total cholesterol (TC), low-density lipoprotein cholesterol (LDL-C), or triglycerides (TG), and decreased high-density lipoprotein cholesterol (HDL-C) [1,2]. This disorder can be classified based on the specific lipid abnormalities, with common types including hypercholesterolemia, hypertriglyceridemia, and mixed dyslipidemia, the latter characterized by increased LDL-C and TG and decreased HDL-C [3,4]. Mixed dyslipidemia is particularly concerning due to its association with an elevated risk of cardiovascular disease [5]. Dyslipidemia can be attributed to genetic factors (primary dyslipidemia) or lifestyle influences (secondary dyslipidemia), with the latter being a significant contributor in 30–40% of cases [6,7].

Dyslipidemia is the most significant risk factor leading to cardiovascular disease (CVD) and its complications [8,9]. Elevated blood lipid levels significantly increased LDL-C, leading to the formation of arterial plaques, causing arterial narrowing and hardening, ultimately resulting in atherosclerosis [2,10]. This progression heightens the risk of atherosclerotic cardiovascular disease (ASCVD) [8,9,11], which poses a severe threat due to potential complications such as stroke, coronary artery disease, and peripheral artery disease [12]. Globally, CVD is the leading cause of death, with two-thirds of cases attributed to ASCVD [2,12]. Individuals with dyslipidemia are twice as likely to develop CVD compared to those with normal lipid levels [3]. Statistics indicate that lipid disorders account for over 4 million CVD deaths annually worldwide, with 3.81 million deaths in 2021 alone linked to uncontrolled LDL-C levels [3,12]. Research suggests that a one mmol/L increase in LDL-C correlates with a 16% rise in CVD incidence [11].

In Asia, the prevalence of dyslipidemia is exceptionally high and continues to grow [13]. Although Asian populations tend to have lower LDL-C levels, elevated TG levels, and low HDL-C levels contribute to a high risk of coronary heart disease [13]. Over the past three decades, CVD-related deaths in Asia have nearly doubled from 5.6 million to 10.8 million [3]. In recent years, a rising incidence of hypercholesterolemia has been noted in countries like Indonesia, Thailand, China, and South Korea, surpassing the prevalence observed in Western nations and Europe [14,15]. This trend is accompanied by an increase in metabolic syndrome cases, including obesity, hypertension, and insulin resistance, further exacerbating the cardiovascular disease burden in the region [15,16,17,18,19].

Thus, reducing LDL-C is the primary goal to control dyslipidemia, thereby preventing the risk of ASCVD [1,20,21]. Statin is the first-line treatment to reduce LDL-C [8,21,22,23]. It is estimated that every 40 mg/dL reduction in LDL-C will reduce the risk of ASCVD by 22% [1,22,24]. The LDL-C reduction mechanism of statins is mainly due to inhibition of the enzyme HMG-CoA reductase, which plays a vital role in cholesterol synthesis [23]. Low doses reduce LDL-C by less than 30%, medium doses reduce LDL-C by 30–50%, and high doses reduce 50%. This classification is based on the degree of LDL-C reduction that corresponds to the patient’s risk of cardiovascular disease [8,24]. Most clinical recommendations support the use of high-dose statins for high-risk subjects (including those with at least some signs of ASCVD [21].

Pitavastatin, the newest class of statins, has drawn a lot of interest lately due to its ability to cut LDL-C as effectively as commonly used statins such as Atorvastatin while also improving HDL-C, therefore reduces the risk of cardiovascular events better, and has a good safety profile [25,26]. The Livalo Effectiveness and Safety (LIVES), a post-marketing surveillance study involving nearly 20,000 pitavastatin participants, reported a very low incidence of statin-induced side effects. Specifically, around 1% of patients experienced myalgia, and only one case of rhabdomyolysis—an infrequent and profound statin-related adverse effect—was documented, suggesting better tolerability compared to other statins [27]. However, clinical recommendations only focus on European and American subjects, so the evidence on statins may not necessarily apply to Asians due to differences in lifestyle, BMI, diet, and genotype [13]. Therefore, this study will summarize the evidence on pitavastatin’s efficacy, effectiveness, and safety in the Asian region.

## 2. Method

### 2.1. Searching Strategy

The study was conducted in October 2024. The search and results presentation are based on the Preferred Reporting Items for Systematic Reviews and Meta-Analyses 2020 (PRISMA) review writing guidelines [28]. With no time limit, extensive databases such as Pubmed, Cochrane, and Embase were searched for pertinent clinical trials and observational research. In which the last publication was retrieved on 22 November 2024. Topic-specific English keywords were typed into the search bar and combined using the Boolean algorithm—AND, OR, or NOT—to form a complete code. Different search syntaxes were developed to ensure the inclusion of all relevant trials presented in Appendix A. The final search query was stated as follows: (pitavastatin) AND ((dyslipidemia) OR (hyperlipidemia) OR (hypercholesterolemia)) AND ((efficacy) OR (effectiveness) OR (safety)). In addition, this review was registered in the Open Science Framework using the following link: https://osf.io/mxkjf (Accessed on 29 November 2024).

### 2.2. Selection Process and Criteria for Research Selection

The Population, Intervention, Comparison, and Outcome (PICO) principle was applied to establish the initial framing of the research question. (1) Participant: patients who are diagnosed with secondary dyslipidemia, at least presenting a high baseline LDL-C over 130 mg/dL; (2) Intervention: pitavastatin as monotherapy; (3) Comparison: other approved statins (also as a single regimen) or different doses of pitavastatin; (4) Outcome: percentage of change in LDL-C.

The trial was deemed eligible if it fulfilled the following requirements: (1) The trial design was either a randomized controlled clinical trial (to establish efficacy) or an observational trial or non-RCT (to analyze effectiveness). (2) The intervention compared statins with each other and must include pitavastatin as single-agent therapy, which could be a fixed or titrated dose. (3) The emphasis of the health condition was on dyslipidemia, with or without cardiovascular disease or diabetes at the same time. (4) The participants were adults. (5) The target subjects are Asian. (6) The study examined the change in LDL-C levels before and after treatment at a given time point. (7) The entire article was accessible, and the presentation was delivered in English.

Otherwise, a trial was considered invalid if it met the following conditions: (1) The experiment lay within a randomized trial scope. However, it was a crossover study, or the article would likely be an individual analysis, such as a case report. (2) A study that did not mention pitavastatin, a study that mentioned pitavastatin, but the dosage was not specified, a study that compared pitavastatin with a non-statin group (e.g., fenofibrate), or a study compared pitavastatin alone versus pitavastatin plus another medication (e.g., ezetimibe). (3) The diagnosis did not center on dyslipidemia; the study was linked to HIV or chronic renal failure, and the investigation concentrated on familial hypercholesterolemia. (4) There was either no discernible assessment time point or insufficient information demonstrating the change in LDL-C levels before and after treatment. (5) Animals were used as subjects for research. (6) The analyzed individuals are not Asian. (7) The entire paper was unreadable, or the study was not in English.

### 2.3. Data Extraction

The following initial information was extracted after the full text was accessed: author, publication year, country, intervention being used and its comparator, medical status of participants, baseline lipid profile, age, sample size (for safety and efficacy), distribution ratio, efficacy outcome, study duration, conduction site. The primary outcome was the efficacy and effectiveness of pitavastatin, which was determined through the percentage change in LDL-C at the endpoint versus baseline. The rate of patients achieving LDL-C goal was extracted if possible. Changes in HDL-C, TC, and TG, among other markers, were also noted.

Our secondary outcome is the safety profile of pitavastatin. This included recording the proportion of patients experiencing adverse events and identifying the most common symptoms associated with pitavastatin. Adverse events were defined as any side effects related to the use of statins during the follow-up period, with serious adverse events also being reviewed. Additionally, parameters of liver function, kidney function, and muscle toxicity were reported.

For liver function, alanine aminotransferase (ALT) and aspartate aminotransferase (AST) levels were analyzed to detect signs of liver damage, with levels three times above the standard threshold (≥×3 ULN) indicating severe liver damage and necessitating discontinuation of the drug. Kidney function was assessed based on plasma creatinine levels, with a level ≥×1.5 ULN indicating impaired kidney function. Muscle damage was monitored through the creatine kinase (CK) index, with levels ten times above the standard threshold (≥×10 ULN) causing myopathy (muscle-related disorder) and requiring cessation of treatment.

### 2.4. Data Synthesis

Data synthesis is performed in a narrative form. The outcome data from each experiment related to lipid profiles are summarized and presented in one comprehensive table. Numerical data are displayed as the mean value, with standard deviation (SD) or standard error (SE) included where possible to illustrate the spread of the data. Pitavastatin is compared to other statins and different doses of pitavastatin. Outcomes are classified based on the dose of pitavastatin to assess its capability to reduce LDL-C.

Regarding efficacy and effectiveness outcome, the research identifies trends and patterns across trials based on the percentage change in LDL-C between specific doses of pitavastatin and other statins. Comparisons are made for short-term (less than one year) and long-term (over one year) durations. Additionally, the study examines whether the findings are consistent under ideal conditions (from randomized controlled trials, RCTs) and in real-world environments (from non-RCTs).

A similar synthesis approach is applied to the safety profile of pitavastatin, with a primary focus on adverse events. The frequency, severity, and discontinuation rates are examined to determine the tolerability of pitavastatin. In particular, the incidence of musculoskeletal symptoms is used as a benchmark to compare the benefit of pitavastatin against other statins. Furthermore, the sample size in each treatment arm is reported to help explain discrepancies in adverse event rates across trials, particularly concerning statin-induced toxicity in different organs (liver, kidney, and muscle).

To ensure the reliability of the findings, all stages—from searching to synthesizing information, were conducted by two independent researchers, with all results approved by both. Following the identification phase, the researchers screened titles and abstracts and performed full-text screening based on inclusion and exclusion criteria. Trials that fulfilled the conditions then proceeded to access the full content. In this stage, crucial data were synthesized and discussed between the two researchers, ultimately leading to the final count of total qualified experiments.

### 2.5. Quality Assessment

#### 2.5.1. For Randomized Controlled Trials

The Consolidated Standards of Reporting Trials (CONSORT) 2010 toolkit was used to evaluate the quality of randomized controlled clinical trials [29]. CONSORT 2010 is a widely used international standard for assessing the quality of clinical trial reporting. The CONSORT 2010 checklist provides a detailed framework for readers to evaluate the reliability and accuracy of a study. To determine an analysis based on CONSORT 2010, people will compare the report with each item in the checklist, from the title, abstract, research methods, results, and discussion. Thereby, readers can evaluate whether the study is scientifically designed, conducted, and reported rigorously. Compliance with the CONSORT 2010 criteria enhances research transparency, improves reproducibility, and provides high-quality scientific evidence to support medical decision-making. This assessment is based on 25 criteria covering the main sections of a clinical trial: title, abstract, introduction, methods, results, and discussion.

First, each study will be compared against the corresponding items to ensure all the crucial data are presented. Then, the researcher will give a certain point, depending on the comprehensiveness of that information. There are three scoring levels: a study is rated as “1” if all required information is provided, “0.5” if the trial partially presents the data, and “0” if no information is fulfilled or the category is not applicable. Each study receives a total score of 25 and a minimum of 0. Studies scoring between 19 and 25 (meeting 75% of the criteria) are considered good quality. Scores ranging from 13 to 18.5 are assessed as moderate quality. Finally, the study is deemed poor quality if the score exceeds 13 (meeting less than 50% of the criteria).

#### 2.5.2. For Non-Randomized-Controlled Trials

The quality of non-randomized-controlled trials (non-RCTs) was evaluated using the Strengthening the Reporting of Observational Studies in Epidemiology (STROBE) scale [30]. STROBE is a widely used international standard for assessing the reporting quality of epidemiological observational studies. The STROBE checklist provides a detailed framework to ensure that observational studies are reported in a complete, transparent, and understandable manner. To evaluate a study based on STROBE, the study report is compared with each item in the checklist, from the title, abstract, and research methods to results and discussion. Researchers can assess whether the study is scientifically designed, conducted, and reported rigorously. Compliance with the STROBE criteria enhances research transparency, improves reproducibility, and provides high-quality scientific evidence to support decision-making in health and related fields. The scale consists of 22 criteria, each scored as follows: 1 if the requirement is fully met, 0.5 if the requirement is partially met, and zero if the requirement is not met. With a maximum possible score of 22, studies scoring between 16 and 22 have low bias, those scoring between 8 and 15 have moderate bias, and studies scoring seven or fewer are considered to have high bias.

### 2.6. Methodological Assessment

To assess the risk of bias, the Joanna Briggs Institute (JBI) appraisal tools for randomized controlled trials (RCTs) [31] and cohort studies [32] were utilized. The JBI checklists are designed to evaluate the methodological quality of trials, helping researchers determine how well the trials were conducted and assess whether their results are trustworthy. For RCTs, the checklist includes 13 questions to ensure each study’s internal validity. For non-RCTs, the cohort checklist consists of 11 questions focusing on the external validity of the studies.

Each question in both checklists is answered as “Yes”, “No”, “Unclear”, or “Not applicable”. A “Yes” answer is awarded 1 point, making the maximum score 13 for RCTs and 11 for non-RCTs. The total score is then translated into an assessment of the risk of bias: “High”, “Moderate”, or “Low”.

For RCTs, a total score of 6 or below is categorized as a “high” risk of bias, a score between 7 and 9 is considered “moderate,” and a score of 10 or above is regarded as a “low” risk of bias. Similarly, for non-RCTs, a score of 5 or below is classified as a “high” risk of bias, a score between 6 and 8 is considered “moderate”, and a score of 9 or above is rated as a “low” risk of bias.

On the other hand, this review does not involve meta-analysis requiring statistical data pooling but rather comparing main findings qualitatively across trials. Hence, the assessment of certainty of evidence was not performed. The transparency of the review is ensured by identifying potential biases that could influence the study results (using the JBI appraisal tool) and by reporting sufficient data for each trial according to the CONSORT and STROBE guidelines. Furthermore, reliable, widely recognized, and accessible online resources such as PubMed, Cochrane, and Embase allow other researchers to easily verify the information and assess the quality of the evidence.

## 3. Results

### 3.1. Selection Process

The screening and selection process for the trials is illustrated in Figure 1. A total of 1137 trials were identified, including 329 from PubMed, 82 from Cochrane, and 722 from Embase, with the most recent trial dated 15 November 2024. After removing 86 duplicates, 1051 studies remained. These studies were then screened by title and abstract according to the inclusion criteria, resulting in 1020 studies being excluded. This left 31 studies that continued to perform full-text screening based on the exclusion criteria. As a result, 14 articles were disqualified, and the total was cut down to 17 articles ready for examining eligibility in full articles. Ultimately, 17 studies were qualified for data synthesis.

### 3.2. Characteristics of Selected Articles

The overall information of the chosen evaluation is demonstrated in Table 1. An estimation of 12/17 studies (70.59%) was RCT [33,34,35,36,37,38,39,40,41,42,43,44], and 5/17 publications were observational studies or non-RCT [45,46,47,48,49]. Most of the trials were conducted in Japan, making up 10/17 studies (58.82%) [34,36,39,41,42,43,44,46,47,48], and the second most common nation was Korea (5/17 trials, 29.41%) [35,37,38,45,49]. The other countries included Thailand [33] and Taiwan [40]. Eligible patients were individuals diagnosed with hypercholesterolemia or hyperlipidemia, which had LDL-C over 130 mg/dL and/or TC more than 200 mg/dL at baseline, except two trials that did not provide information in terms of initial TC [35,42].

On the other hand, the majority of publications focused on analyzing pitavastatin compared to other statins, comprising 13/17 articles [33,34,35,36,37,38,39,40,41,42,44,47,49]. In which the most common comparators were atorvastatin, accounting for 11/13 trials [33,34,35,38,39,40,41,42,44,47,49], followed by rosuvastatin [34,49], simvastatin [37], and pravastatin [36,49]. The rest of the studies compared the efficacy or effectiveness among different doses of pitavastatin [43,45,46,48]. All of the statin doses were set into one-time usage per day. In addition, the number of participants fluctuated from 100 to 600 people for a large portion of studies (12/17 trials) [34,35,36,38,40,41,42,43,44,46,47,49]. A total of 4/17 articles had no more than 100 persons [33,37,39,48] and 1 was reported to have the most significant number of participants, with more than 27,000 people [45]. Most of the trials were held in hospitals, clinical centers, and institutes; 4/16 articles did not mention any data about this [43,46,48,49].

### 3.3. Quality Assessment

#### 3.3.1. CONSORT 2010 Tool for RCT

The result of the quality assessment of included RCT is summarized in Appendix A. About 66.67% of the articles (8/12 RCT) were deemed as good quality [33,34,35,37,38,40,42,44], with the total score fluctuating from 20 to 23.5. The rest of the trials (accounted for 33.33%) were graded as moderate quality, ranging from 13 to 18.5 [36,39,41,43].

#### 3.3.2. STROBE 2014 Tool for Non-RCT

For non-RCT, the quality of each trial through the STROBE 2014 checklist is presented in Appendix A. It was estimated that 100% of trials portrayed a low risk of bias, with the total score ranging from 17 to 20.5.

### 3.4. Methodological Assessment

The methodological quality of RCT and non-RCT following the JBI appraisal tool is summarized in Appendix A, respectively. For RCT, nine out of twelve trials (accounted for 75%) were graded as low risk of bias [33,34,36,37,40,41,42,43,44], with the total score ranging from 9 to 11. The rest was concluded as moderate methodological quality, which scored 9/13 [35,38,39]. Notably, most experiments failed to illustrate the evidence within question 4: “Were target subjects blind to treatment assignment?” and question 5: “Were blinding mentioned in those delivering the treatment assignment?”. Furthermore, most RCTs did not provide sufficient evidence related to the blinding of outcome assessors. They thus were graded as “unclear” in question 7: “Were outcome assessors or investigators blind to treatment assignment?”.

On the other hand, four out of five observational studies (made up 80%) portrayed straightforward methodological procedure, which was graded as low risk of bias. All of these experiments had a total score of 9/11 [45,47,48,49]. The other one left scored 7/11, which was graded as moderate [46]. In addition, all five trials were graded “unclear” regarding question 10: “Were methods used to deal with insufficient follow-up recorded?”.

### 3.5. Efficacy, Effectiveness

#### 3.5.1. Between Pitavastatin and Other Statins

The percentage of mean change in lipid profile between pitavastatin and other statins is illustrated in Table 2. Pitavastatin was proven to reduce significantly LDL-C from 28% to 42.9% after 8 weeks to 30 months. Notably, there was a change in LDL-C reduction efficacy when pitavastatin was compared to different doses of atorvastatin.

At a low dose, pitavastatin 1 mg was estimated to lower LDL-C by 37–38% after 2–3 months. In comparison to atorvastatin 10 mg, the RCT result showed that PIT 1 mg provided a significantly weaker LDL-C reduction effect within 3 months (−37.37% vs. −45.57, *p* = 0.005) [33]. Nevertheless, an observational trial in Japan concluded that the effectiveness was the same between these two interventions but not statistically different after 2 months [47].

When increasing the potency to 2 mg, PIT could decrease 35–42.9% of LDL-C in the short term (2–4 months). This efficacy was reported to be similar to atorvastatin 10 mg, mentioned in 3/4 RCT [38,40,41]. In which an insignificant difference was found in 1 RCT [38]. At the same time, a retrospective study in Korea also portrayed consistent results, with the % change in LDL-C in PIT vs. ATOR was −41.7% vs. −44.1%, *p* < 0.05, respectively [49]. Regarding long-term treatment (>1 year), LDL-C reduction in PIT 2 mg was around 33–36% and could not be concluded compared to ATOR due to inconsistent results between 2 RCTs in Japan. According to Sasaki et al. 2008, PIT 2 mg was less effective significantly than ATOR 10 mg in terms of LDL-C reduction effect (−33.0% vs. −40.1%, *p* = 0.002) [42]. However, Moroi et al. found that their efficacy was equivalent and did not demonstrate a significant difference (−36% vs. −37%, *p* > 0.05) [44].

The moderate–high dose, on the other hand, demonstrated that PIT 2–4 mg could alleviate LDL-C approximately 34.6–41% after 3–4 months and dropped to 28.7% after 2.5 years. This pattern was similar to ATOR 10–20 mg both in short-term and long-term treatment, as shown in 3 RCTs [34,35,39]. Of these, 2/3 of RCTs demonstrated no significant difference, with one in a 2.5-year-length [39] and one in the 4-month-length trial [34].

Compared to rosuvastatin, the efficacy difference was observed when changing dose potency in PIT, 1 RCT, and one non-RCT. Within RCT, the author estimated a similar efficacy between PIT 2–4 mg and ROS 2.5–5 mg after 4 months, but not significantly different (−41% vs. −42%, *p* > 0.05) [34]. Additionally, an observational study in Korea reported a similar range effect but at different doses. Their finding illustrated that a fixed dose of PIT 2 mg/day reduced LDL-C weaker than ROS 5 mg and ROS 10 mg, with a significant difference found [49]. Moreover, PIT 2 mg was as effective as Simvastatin 20 mg after 2 months, with no significant difference (*p* = 0.648) [37]. Compared to PRA, PIT 2 mg was two times more potent than PRA 10 mg (−37.6% vs. −18.4%, *p* = 0.001) [36] but showed a similar percentage significantly to PRA 40 in terms of LDL-C reduction capability (−41.7% vs. −42.4%, *p* < 0.005) within 3–6 months [49].

Regarding the rate of patients achieving the LDL-C target, the rate was high and similar between PIT and other statin such as ATOR [33,34,38,40,49], ROS [34,49] and SIM [37] within 2–6 months of treatment. It should be noted that the majority of trials reported no significant difference [33,37,38,40]. However, compared to PRA 10 mg, PIT 2 mg showed a double rate in terms of people successfully attaining LDL-C target [36]. Still, compared to PRA 40 mg, PIT 2 mg demonstrated an equivalent LDL-C achievement rate; both groups were over 90% [49]. Most of the trials found that the LDL-C attain rate in the PIT group was around 75–95%, except for one trial in Taiwan in 2013 (the incidence of patients achieving the LDL-C goal was 50%) [40].

#### 3.5.2. Between Different Doses of Pitavastatin

Efficacy and effectiveness at different doses of pitavastatin are illustrated in Table 3. All four trials showed pitavastatin effectively reduced LDL-C from 2 months to a year. The result was consistent between RCT in Japan [43] and non-RCT in Korea [45], indicating a dose-dependent trend when analyzing the capability of LDL-C reduction between PIT 1 mg vs. PIT 2 mg vs. PIT 4 mg in patients not treated with lipid-lowering medication. As the PIT dose increased, the LDL-C-reducing effect became stronger. At the lowest dose, PIT 1 mg/day showed 26–33% LDL-C reduction; the rate continued to rise to 40% at 2 mg per day and reached its peak at 46–47% at the maximum dose of 4 mg/day [43,45,48].

### 3.6. Safety

#### 3.6.1. Adverse Drug-Related Event

Detailed information regarding the drug-related adverse event, serious adverse event, and tolerability of pitavastatin compared to other statin groups are shown in Table 4. Using pitavastatin at 1 mg, 2 mg, or 4 mg per day was proven safe and well-tolerated in all of the studies during the treatment period (from 2 months to 2.5 years). Incidence of adverse drug reactions was low (less than 20%) and was reported to be comparable between PIT 1–2 mg and other statins: atorvastatin 10 mg–20 mg, rosuvastatin 2.5–5 mg, pravastatin 10 mg, simvastatin 20 mg. Notably, there was no significant difference in most of the studies (*p* < 0.05).

The most common adverse event associated with pitavastatin included gastrointestinal disorders (vomiting, stomach ache, abdominal pain, indigestion, diarrhea) [33,34,35,37,38,42,43], symptoms on the central nervous system (dizziness, headache, anxiety) [35,38,42], skin disorders (edema, rash, dermatitis, pruritus) [34,35,39,40,42,43], upper respiratory problems (cough, nasopharyngitis) [35,40] or general disorder such as fatigue, malaise [34,39,42,43].

Additionally, pitavastatin was estimated to cause side effects on musculoskeletal sites, including myalgia [33,34,37,38,40,42,45], muscle spasms [33,34], and muscle stiffness [34]. Even though the chance that patients encountered myalgia was comparably low (around 1%), the pitavastatin 1–2 mg group was recorded to have a higher incidence than atorvastatin 10 mg [33,34,38,42], rosuvastatin 2.5–5 mg [34]. However, all of the adverse events were mild to moderate in severity. Aside from 1 RCT reporting one rhabdomyolysis case in the PIT 2 mg group but none in ATOR 10 mg groups (0.3% vs. 0%, *p* = 1.000) [44], most of the trials did not show any serious adverse events.

#### 3.6.2. Abnormality of the Liver, Kidney Function, and Signs of Muscle Toxicity

The results of liver and kidney function and signs of myopathy associated with pitavastatin are summarized in Table 5. Regarding liver function, pitavastatin was observed to cause an increase in liver enzymes (AST and ALT) but at a low incidence rate. In an observational study involving 28,343 participants, the proportion of patients experiencing liver abnormalities with pitavastatin at 1 mg, 2 mg, and 4 mg was 0.16% over an 8-week evaluation period [45]. The rate of liver enzyme elevation with pitavastatin 2–4 mg was lower than that observed with atorvastatin 10–20 mg, as demonstrated in 4 RCTs [34,38,40,41]. Compared to ROS 2.5–5 mg, the rate of patients experiencing ALT elevation was significantly higher in the PIT 2–4 mg group (4.2% vs. 3.1%, *p* = 0.043), following 4-month-in length RCT [34]. This was contrasted in a non-RCT study analyzing PIT 2 mg vs. ROS 5 mg vs. ROS 10 mg with a total of 355 participants [49]. PIT 2 mg did not record any elevated ALT or AST cases; meanwhile, ROS 5 mg and ROS 10 mg groups revealed 1 patient in each group [49].

The majority of trials reported no cases of AST or ALT elevations greater than three times the standard upper limit (ULN) with pitavastatin use [33,37,38,43,47]. Four studies reported ALT levels exceeding ×3 ULN, with rates ranging from 0.8% to 4.0% after two months of treatment [33,36,42,45]. Among these, only one RCT by Saito et al. found a statistically significant difference, where pitavastatin 2 mg showed a higher risk of ALT elevation compared to pravastatin 10 mg (1.6% vs. 0.9%, *p* < 0.05) after 12 weeks of observation [36].

Pitavastatin use has also been associated with increased creatine kinase (CK) levels, which may indicate myotoxicity [35,36,37,38,43,45,47]. However, the incidence of CK elevation was low, ranging from 0.6% to 5.4% in the pitavastatin group, and no trials reported myopathy (CK > ×10 ULN). Compared to atorvastatin 10–20 mg, pitavastatin 2–4 mg showed similar rates of CK elevation, but the difference was not statistically significant [34,35,41,44]. The proportion of patients with CK elevation was twice as high for pitavastatin 2 mg compared to pravastatin 10 mg (4% vs. 2%) [36] and half that of simvastatin 20 mg (3.8% vs. 9.8%) [37]. Regarding renal function, most trials reported plasma creatinine levels below ×1.5 ULN except one RCT (insignificant difference, *p* = 0.390) [44], indicating no renal damage associated with pitavastatin use.

The majority of trials reported no cases of AST or ALT elevations greater than three times the standard upper limit (ULN) with pitavastatin use [33,37,38,43,47]. Four studies reported ALT levels exceeding ×3 ULN, with rates ranging from 0.8% to 4.0% after two months of treatment [33,36,42,45]. Among these, only 1 RCT by Saito et al. found a statistically significant difference, where pitavastatin 2 mg showed a higher risk of ALT elevation compared to pravastatin 10 mg (1.6% vs. 0.9%, *p* < 0.05) after 12 weeks of observation [36].

Pitavastatin use has also been associated with increased creatine kinase (CK) levels, which may indicate myotoxicity [35,36,37,38,43,45,47]. However, the incidence of CK elevation was low, ranging from 0.6% to 5.4% in the pitavastatin group, and no trials reported myopathy (CK > ×10 ULN). Compared to atorvastatin 10–20 mg, pitavastatin 2–4 mg showed similar rates of CK elevation, but the difference was not statistically significant [34,35,41,44]. The proportion of patients with CK elevation was twice as high for pitavastatin 2 mg compared to pravastatin 10 mg (4% vs. 2%) [36] and half that of simvastatin 20 mg (3.8% vs. 9.8%) [37]. Regarding renal function, most trials reported plasma creatinine levels below ×1.5 ULN except one RCT (insignificant difference, *p* = 0.390) [44], indicating no renal damage associated with pitavastatin use.

## 4. Discussion

### 4.1. Efficacy and Effectiveness

According to our findings, pitavastatin is an effective management for dyslipidemia both in the short term (2 months to less than 1 year) and the long term (>1 year). After 2–4 months of treatment, a dose of 1 mg reduces LDL-C by approximately 37–38%; a dose of 2 mg reduces LDL-C by 35–43%; and a dose of 2–4 mg lowers LDL-C by 34% to 41%. When administered for over a year, a fixed dose of PIT 2 mg diminishes LDL-C by 33–36%, whereas a non-fixed dose of PIT 2–4 mg limits LDL-C by around 28%. The capability for regulating LDL-C is dose-dependent, with higher doses yielding more significant LDL-C reduction, especially in patients using lipid-lowering therapy for the first time. Only two trials employed the lowest dose of PIT, comparing PIT 1 mg to ATOR 10 mg. While the results were conflicting (RCT highlighted reduced efficacy in the PIT group, observational trials revealed similar effectiveness), both demonstrated an LDL-C reduction of 30–40%. This reduction effect is consistent with observational trials evaluating the efficacy of pitavastatin doses ranging from 1 to 4 mg [43,45,46]. This pointed out that the lowest dose of pitavastatin can reduce LDL-C to the same level as the moderate-intensity statin group, which satisfies the ideal LDL-C reduction level of a statin as recommended by NCEP-ATP III [33].

On the other hand, using a fixed dose of PIT 2 mg/day is as effective as ATOR 10 mg/day regarding LDL-C reduction within 2–4 months of treatment. However, the results are still inconsistent in long-term therapy. Depending on the dosage, the efficacy varies in comparison to rosuvastatin. According to an RCT result, PIT 2–4 mg and ROS 2–5 mg were equally effective in reducing LDL-C (40% vs. 42%, *p* > 0.05) [34]. This similar proportion was observed in PIT 2 mg, following a non-RCT in 6 months, but it was lower significantly than ROS 5 mg and ROS 10 mg groups by 10% (*p* < 0.05) [49].

The potent efficacy of pitavastatin in lowering LDL-C is attributed to its high affinity for HMG-CoA reductase and its strong liver selectivity [50]. The unique cyclopropyl group within the structure of pitavastatin enhances its binding to HMG-CoA reductase and shields it from metabolism by cytochrome P450 enzymes, particularly CYP3A4, which is responsible for the metabolism of numerous statins, including simvastatin and atorvastatin [51]. As a result, bioavailability for many statins was significantly reduced, ranging from approximately 5% for simvastatin and lovastatin to 12% for atorvastatin and up to 20% for rosuvastatin [52]. In contrast, pitavastatin’s minimal metabolism by CYP450 enzymes allows it to achieve a bioavailability of 60–80% [52,53]. Patients with dyslipidemia often require multiple medications due to comorbid conditions like diabetes and hypertension. Consequently, the trend in treatment is to prioritize fixed-dose regimens (one-time usage per day) to simplify medication regimens and improve patient adherence [14]. The reduced metabolism by CYP450 also minimizes drug interactions, making pitavastatin a convenient option for combination therapy with other drug classes [54].

Although rosuvastatin and atorvastatin are the two most potent statins for lowering LDL-C [55], atorvastatin is the most commonly prescribed [56,57]. Pitavastatin demonstrates comparable efficacy in lowering LDL-C, suggesting it is a viable alternative when switching medications is necessary. Moreover, pitavastatin’s relatively long half-life of 12 h, which is close to the 15–30 h half-life of atorvastatin and rosuvastatin, enables once-daily dosing [52].

### 4.2. Safety

The adverse effects of statin use on muscle and liver function have been well-documented in numerous clinical trials [58,59]. Hepatic dysfunction, often evident through elevated levels of alanine aminotransferase (ALT) and/or aspartate aminotransferase (AST), is a common consequence of statin therapy [57]. Our analysis confirms that pitavastatin (PIT) is associated with increased AST and ALT levels after 2–3 months of treatment, consistent with findings from several clinical trials. Pitavastatin demonstrated a lower incidence of elevated liver enzymes than atorvastatin, showing comparable results to rosuvastatin. In case either of the values surpasses the ×3 ULN, statin discontinuation is recommended, and patients are advised to proceed with liver-recovery treatment [57]. However, the incidence of patients achieving AST/ALT > ×3 ULN was low in our research. Hepatic dysfunction was primarily indicated by ALT levels exceeding ×3 ULN, with prevalence rates ranging from 0.8% to 4.0% in RCTs [33,35,36,42], and an estimated 0.12% in a Korean observational study [45]. The variation in results between RCTs and non-RCTs can be attributed to differences in sample sizes, with RCTs typically assessing 50 to 600 participants, compared to more than 28,000 individuals in non-RCTs. Based on prior research, the incidence of patients experiencing ALT > ×3 ULN is estimated to be 0.9% [57].

On the other hand, musculoskeletal symptoms are frequently encountered with statin therapy, with muscle pain (myalgia) being the most commonly reported side effect. The underlying mechanism for this toxicity is believed to be related to mitochondrial dysfunction. In addition to its role in cholesterol synthesis, HMG-CoA reductase is involved in the production of Coenzyme Q10 (CoQ10), which is essential for ATP production in mitochondria. A deficiency in CoQ10 impairs mitochondrial energy production, leading to insufficient energy for muscle cells and resulting in muscle damage [60,61]. Pitavastatin is not exempt from this effect; cases of muscle pain and weakness have been reported, though they tend to be mild to moderate in severity, with an incidence ranging from 0.6% to 5.4%. In addition, statins can also cause inflammatory reactions in muscles. Increased creatine kinase (CK) is a sign of muscle damage and is often observed in people treated with statins [50,62]. In our findings, pitavastatin increased CK levels but did not exceed the threshold for myopathy (CK > ×10 ULN). The frequency of CK elevation in patients treated with PIT was similar to ATOR and ROS (no statistical difference) but approximately twice as high as that seen with pravastatin. This discrepancy may be attributed to the solubility of the statins. Hypothetically, hydrophilic (water-soluble) statins, such as rosuvastatin and pravastatin, penetrate muscle cell membranes less effectively than lipophilic (lipid-soluble) statins, like pitavastatin, atorvastatin, and simvastatin, suggesting that pravastatin carries a lower risk of muscle toxicity than pitavastatin [63,64].

The incidence of serious side effect related to statin is relatively rare, less than 0.1% (including rhabdomyolysis) [65,66]. In standard-dose therapy, the incidence of statin-induced rhabdomyolysis is around 3.4 per 100,000 person-years [67]. Following the USA estimate, to have one case of rhabdomyolysis, 22,727 cases of statin monotherapy are needed [67]. However, in our study, we observed only 1 RCT that recorded cases of rhabdomyolysis, which fell into the pitavastatin group of patients (rate of 0.3%), while the group using ATOR had no records of serious adverse events. This can be explained by the small sample size of most of the trials in the study, as only 2/17 trials had a sample size of over 600 people [44,45]. In the Korean trials involving 29,000 participants, no rhabdomyolysis was reported with dominance in pitavastatin 2 mg [45], suggesting pitavastatin is safe, with a very low risk of serious statin-induced myotoxicity. A 2022 study analyzing the risk of rhabdomyolysis across statins, using data from the World Health Organization (WHO) pharmacovigilance system, found that, of 10,657 reported cases of rhabdomyolysis, pitavastatin had the lowest incidence (1.5%), while the highest rates were observed with simvastatin, atorvastatin, and rosuvastatin (40.8%, 31.39%, and 16.67%, respectively) [58]. Collectively, these findings support the conclusion that pitavastatin is a relatively safe and well-tolerated option, with minimal risk of liver, muscle, or kidney damage, and may serve as a viable alternative to the more widely used atorvastatin and rosuvastatin.

### 4.3. Focus on Asia

There is a difference in the initial statin dose prescribed between Asian and Western populations. While international guidelines typically recommend the highest tolerated statin dose to reach LDL-C targets, Asian clinical practice favors starting with lower doses [68]. This approach is common in Japan, China, and India [68,69,70]. Several factors explain this preference. Most importantly, Asian populations show greater sensitivity to statins compared to Western populations [71]. Their higher plasma statin levels increase the likelihood of adverse reactions—especially muscle-related side effects—making high doses harder to tolerate [70,71]. For this reason, both Health Canada and the FDA advise starting Asian patients on lower statin doses to reduce the risk of statin-related myopathy [71].

Genetic polymorphisms also contribute significantly to the statin-induced side effects. All statins (including pitavastatin) are transported to hepatocytes, where they inhibit cholesterol synthesis, via a transporter called organic anion transporting polypeptide (OATP), specifically OATP1B1 [72,73]. The SLCO1B1 gene encodes this protein [72,74]. Variants in SLCO1B1 can decrease the function of this transporter, leading to higher-than-necessary statin concentrations in the bloodstream, especially in muscle tissue. Consequently, this increased exposure can elevate the prevalence of statin-related myotoxicity [75,76,77]. Observational studies have shown that Filipinos and Koreans are at the highest risk of carrying genetic variants linked to an increased risk of myopathy when using rosuvastatin or fluvastatin [78]. Specifically, Filipinos are more likely to experience myopathy when prescribed rosuvastatin, suggesting that dose adjustments may be necessary in these populations [78]. As a result, physicians often prescribe lower statin doses to minimize side effects.

Another contributing factor is the difference in dyslipidemia patterns in Asia. A cross-sectional study analyzing lipid profiles in seven Asian American minority groups—including Asian Indian, Chinese, Filipino, Japanese, Korean, and Vietnamese—compared to non-white populations found that Asians are more likely to have hypertriglyceridemia and low HDL-C, particularly in Asian Indians [79]. This pattern is likely influenced by dietary habits, including the high consumption of carbohydrates and saturated fats, especially in South Asia [18]. Additionally, men in Asian populations are more likely to develop lipid abnormalities than women, with Filipino (73.1%) and Vietnamese men (71.3%) having the highest incidence of increased LDL-C [79]. Historically, Japanese, South Korean, and Chinese populations tend to have lower LDL-C levels than other groups [69,79]. These factors demonstrate the limited need for high-dose statin therapy. Even when the dose is doubled, the additional LDL-C reduction only ranges from 5 to 7%, further supporting that high-intensity statin therapy may not be the most effective option for the Asian population [71]. For example, in Japan, only 3% of patients with acute coronary syndrome are prescribed 4 mg of pitavastatin [69]. Similarly, in China, moderate statin doses are recommended as initial therapy, with pitavastatin starting at 1 mg [70]. These findings further support the idea that higher statin doses are often unnecessary for effective LDL-C reduction in Asian populations.

### 4.4. Strength and Limitation

In contrast to the most recent review by Zhang et al. in 2020, which primarily focused on comparing the LDL-C-lowering effects of seven different statin groups [51], our study specifically assessed the efficacy and safety profile of pitavastatin alone. Unlike their findings indicating that atorvastatin and rosuvastatin were the most potent LDL-C-lowering agents, respectively, with pitavastatin ranking third, our study concluded that pitavastatin 2 mg was equally effective as both ATOR 10 mg, pitavastatin 2–4 mg was similar in efficacy to rosuvastatin 2.5–5 mg in short-term treatment. However, pitavastatin was less effective at doses of 2 mg than rosuvastatin 5 mg and 10 mg.

Most of the clinical trials in our study focused on comparisons between pitavastatin and atorvastatin—the most popular drug on the market, providing data that closely aligns with real-world prescribing practices. This study has added an alternative option for doctors and health workers to use flexibly in addition to atorvastatin and rosuvastatin. This is also the first study focusing on Asian subjects, filling the gap between international recommendations, concentrating only on Europeans and Americans. In another observational study, the author concluded that Asian people (especially in the South), such as India, show a stronger response to statin therapy, achieving significant LDL-C reduction with lower doses. However, this group may also be at a higher risk of experiencing side effects [8].

However, this study has several limitations. The study was only conducted in a limited scope based on three popular databases: PubMed, Cochrane, and Embase, so it is inevitable that there is a lack of some studies in other databases. The sample sizes in most of the RCTs included were relatively small, typically ranging from 100 to 600 participants, which made it challenging to accurately estimate the proportion of patients who experienced severe side effects from pitavastatin. As a result, these figures may not fully reflect the actual experiences. Additionally, while changes in lipid parameters such as HDL-C, total cholesterol, and triglycerides were observed, a more detailed discussion of these parameters was not conducted. Although a slight improvement in HDL-C was presented, 80% of the trials (12/17 articles) recorded normal baseline HDL-C levels (>50 mg/dL) in dyslipidemia patients [33,34,36,37,38,41,42,43,44,45,46,47,48], which suggests that the potential of pitavastatin to enhance HDL-C levels has not been fully explored.

## 5. Conclusions

Pitavastatin at 1 mg, 2 mg, and 4 mg effectively treats dyslipidemia by significantly reducing LDL-C levels after at least two months of treatment. The higher the dose, the more significant the reduction in LDL. The lowest dose of pitavastatin can reduce LDL-C by 37% to 38%, comparable to statins with moderate intensity. A dose of 2 mg can reduce LDL-C by approximately 33% to 43%, similar to the efficacy of atorvastatin 10 mg in short-term treatment (2–4 months), although its long-term effectiveness (over 1 year) remains uncertain. Pitavastatin at 2–4 mg is as effective in lowering LDL-C as rosuvastatin at 2.5–5 mg. However, pitavastatin is less effective at a fixed dose of 2 mg than rosuvastatin, 5 mg, and 10 mg. Compared to less commonly used statins, pitavastatin 2 mg is more effective than pravastatin 10 mg and equivalent to pravastatin 40 mg and simvastatin 20 mg after 2–3 months of treatment.

Adverse effects on the musculoskeletal system, such as myalgia, were observed with pitavastatin use, but the incidence was very low, and the symptoms were mild. Biochemical parameters indicated increases in liver enzymes (AST, ALT) and creatine kinase levels, suggesting muscle toxicity potentially, but these increases did not exceed the allowable thresholds. Most trials did not report any serious adverse effects, such as rhabdomyolysis. Overall, pitavastatin was well tolerated and safe throughout the treatment period.

## 6. Future Directions

Pitavastatin, a potent statin, is poised for increased utilization in the future due to several factors. Its high efficacy in lowering LDL-C with a relatively low risk of drug interactions and myopathy makes it an attractive option, particularly for patients on complex medication regimens or those intolerant to other statins. Ongoing research exploring the pleiotropic effects of pitavastatin, such as its anti-inflammatory and antioxidant properties, may further expand its therapeutic applications beyond dyslipidemia, potentially in cardiovascular disease prevention and management. Additionally, cost-effectiveness analyses supporting pitavastatin’s inclusion in formularies and guidelines may improve patient access and contribute to its broader adoption in clinical practice. This suggests a promising future for pitavastatin in addressing the global burden of cardiovascular disease.

## Figures and Tables

**Figure 1 healthcare-13-00059-f001:**
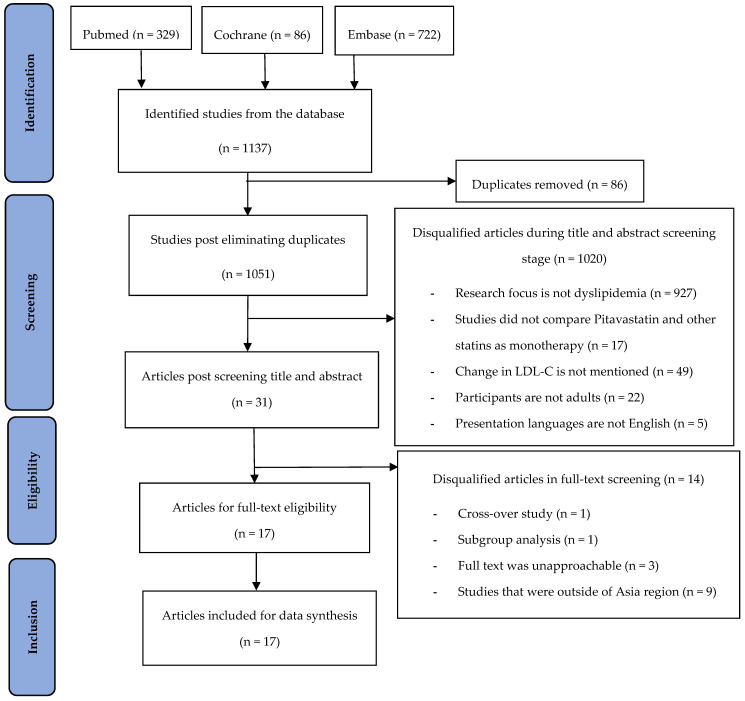
PRISMA flow diagram for identifying and selecting articles.

**Table 1 healthcare-13-00059-t001:** Overall features of included studies.

No.	Author, Year, Ref	Country	Study Design	Intervention (mg/day)	Medical Status	Mean Baseline (mg/dL)	Age (Years)	Efficacy Sample	Safety Sample	Ratio	Outcome	Conduction Period	Conduction Site
1	Jeong et al., 2020 [45]	Korea	Non-RCT	PIT 1 vs. PIT 2 vs. PIT 4	Hyperlipidemia	200 ≤ TC ≤ 220 LDL-C ≥ 130 150 ≤ TG < 400	≥20	27,901	28,343	-	% LDL-C reduction	4/2012–4/2017	893 facilities
2	Koshiyama et al., 2008 [46]	Japan	Non-RCT	PIT 1-2	Hypercholesterolemia	TC ≥ 220 LDL-C ≥ 130 150 ≤ TG < 400	62.0 ± 0.9 ^S.E^	178	178	-	% LDL-C reduction	-	-
3	Yamasaki et al., 2006 [48]	Japan	Non-RCT	PIT 1 vs. PIT 4	Dyslipidemia, hypertension	TC ≥ 220 LDL-C ≥ 130 TG ≤ 150	≥20	63	63	1:1	% LDL-C reduction	-	-
4	Kong et al., 2017 [49]	Korea	Non-RCT	PIT 2 vs. ATOR 10,20 vs. ROS 5,10 vs. PRA 40	Dyslipidemia, TD2M	180 ≤ TC ≤ 220 LDL-C ≥ 100 TG ≤ 150 HDL-C ≤ 50	40–75	355	355	-	% LDL-C reduction	-	-
5	Yoshitomi et al., 2006 [47]	Japan	Non-RCT	PIT 1 vs. ATOR 10	Hypercholesterolemia	TC ≥ 220 LDL-C ≥ 130 150 ≤ TG < 400	≥18	137	137	1:1	% LDL-C reduction	12/2002 (ATOR), 9/2003–1/2005 (PIT)	5 primary care centers in Shizuoka
6	Sansanay udh et al., 2010 [33]	Thailand	RCT	PIT 1 mg vs. ATOR 10 mg	Hypercholesterolemia	TC ≥ 220 LDL-C ≥ 130 TG ≤ 150	≥18	98	98	1:1	% LDL-C reduction	11/2008–5/2009	Phramongkutklao hospital, Bangkok
7	Saku et al., 2011 [34]	Japan	RCT	PIT 2–4 vs. ATOR 10–20 vs. ROS 2.5–5	Hypercholesterolemia, at-risk CHD patients	LDL-C ≥ 140 TG ≤ 150	20–75	285	298	1:1:1	% LDL-C reduction	-	Fukuoka University Hospital and 50 other related ones in Kyushu
8	Han et al., 2012 [35]	Korea	RCT	PIT 2–4 vs. ATOR 10–20	Hypercholesterolemia with elevated ALT	×1.25 ULN ≤ ALT ≤ ×2.5 ULN TC ≥ 220 LDL-C ≥ 130 150 ≤ TG < 400 HDL-C ≤ 50	25–75	173	189	1:1	% LDL-C reduction		10 hospitals
9	Saito et al., 2002 [36]	Japan	RCT	PIT 2 vs. PRA 10	Hypercholesterolemia	TC ≥ 220 LDL-C ≥ 130 150 ≤ TG < 400	20–75	236	233	1:1	% LDL-C reduction	-	43 institutes
10	Park et al., 2005 [37]	Korea	RCT	PIT 2 vs. SIM 20	Hypercholesterolemia	TC ≥ 220 LDL-C ≥ 130 150 ≤ TG < 400	20–75	95	103	1:1	% LDL-C reduction	10/2003–7/2004	Yonsei University College of Medicine, Seol National University Hospital
11	Lee et al., 2007 [38]	Korea	RCT	PIT 2 vs. ATOR 10	Hypercholesterolemia	TC ≥ 220 LDL-C ≥ 130 TG ≤ 150	20–79	222	268	1:1	% LDL-C reduction	5/2005–1/2006	18 clinical centers
12	Moroi et al., 2020 [44]	Japan	RCT	PIT 2 vs. ATOR 10	Hypercholesterolemia, ≥1 risk factor for ASCVD	TC ≥ 220 LDL-C ≥ 140 150 ≤ TG < 400	≥35	622	622	1:1	% LDL-C reduction	04/2006–05/2011	3 medical centers of Toho University
13	Kurogi et al., 2013 [39]	Japan	RCT	PIT 2–4 vs. ATOR 10–20	Hypercholesterolemia reduced HDL-C, stable CAD	TC ≥ 190 LDL-C ≥ 100 HDL-C ≤ 50 TG < 150	20–85	71	71	1:1	% LDL-C reduction	9/2008–9/2009	17 hospitals
14	Liu et al., 2013 [40]	Taiwan	RCT	PIT 2 vs. ATOR 10	Hypercholesterolemia, TD2M (with or without), high-risk patients based on NCEP-ATP III criteria	TC ≥ 200 LDL-C ≥ 130 150 ≤ TG < 400 HDL-C ≤ 50	≥20	194	225	1:1	% LDL-C reduction	7/2011–4/2012	6 medical centers
15	Yokote et al., 2008 [41]	Japan	RCT	PIT 2 vs. ATOR 10	Hypercholesterolemia including FH	TC ≥ 220 LDL-C ≥ 130 TG < 400	≥20	191	-	1:1	% LDL-C reduction	1/2005–4/2005	39 primary care and specialist centers in Chiba, Tochigi, Kanagawa and Shizuoka prefectures
16	Sasaki et al., 2008 [42]	Japan	RCT	PIT 2 vs. ATOR 10	Elevated LDL-C, glucose intolerance	LDL-C ≥ 140 150 ≤ TG < 500	≥20	173	189	1:1	% LDL-C reduction	10/2004–3/2007	34 clinics, hospital in Kyushu Island
17	Saito et al., 2011 [43]	Japan	RCT	PIT 1 vs. PIT 2 vs. PIT 4	Hyperlipidemia	TC ≥ 220 LDL-C ≥ 130 150 ≤ TG < 400	25–75	266	260	1:1:1	% LDL-C reduction	-	-

Abbreviation: TD2M: type 2 diabetes mellitus; FH: familial hypercholesterolemia; CAD: coronary artery disease; CHD: coronary heart disease; ASCVD: atherosclerotic cardiovascular disease; HDL-C: high-density lipoprotein cholesterol; LDL-C: low-density lipoprotein cholesterol; TC: total cholesterol; TG: triglyceride; PIT: pitavastatin; ATOR: atorvastatin; ROS: rosuvastatin; PRA: pravastatin; SIM: simvastatin; ALT: alanine aminotransferase. ^S.E^: standard error.

**Table 2 healthcare-13-00059-t002:** Mean change in lipid profile between pitavastatin and another statin group.

Author, Country, Year	Endpoint	LDL-C			HDL-C (%)	TC (%)	TG (%)	Conclusion Focusing on LDL-C Reduction
		Mean Baseline ± SD (mg/dL)	Change (%)	Patients Achieving Goal (%)				
Pitavastatin 1 mg vs. Atorvastatin 10 mg								
Yoshitomi et al., Japan 2006 ^Non-RCT^ [47]	12 weeks	182 ± 28 188 ± 28	−38 ± 13 −41 ± 12 *p* > 0.05	- - -	3 ± 12 7 ± 12 *p* > 0.05	−28 ± 8 −29 ± 10 *p* > 0.05	−11 ± 30 −21 ± 25 *p* < 0.05	PIT 1 mg/day was as effective as ATOR 10 mg/day
Sansanayudh et al., Thailand 2010 [33]	8 weeks	175.99 ± 34.54 172.86 ± 34.53	−37.37 ± 11.37 −45.75 ± 10.60 *p* = 0.005	74 ^t1^ 84 ^t1^ *p* = 0.220	2.76 ± 17.94 −0.41 ± 11.41 *p* = 0.294	−27.55 ± 8.06 −32.31 ± 8.37 *p* = 0.005	−10.37 ± 38.92 −7.06 ± 36.33 *p* = 0.661	PIT 1 mg/day was weaker than ATOR 10 mg/day
Pitavastatin 2 mg vs. Atorvastatin 10 mg								
Kong et al., Korea 2017 ^Non-RCT^ [49]	6 months	121.6 ± 20.8 134.0 ± 28.2	–41.7 −44.1 *p* < 0.05	96.5 ^t2^ 81.7 ^t2^	–1.6 0.0	–15.2 ± 7.9 –24.7 ± 13.4	–15.9 −15.5	PIT 2 mg/day was as effective as ATOR 10 mg/day
Lee et al., Korea 2007 [38]	8 weeks	159 ± 21 160 ± 23	−42.9 ± 12.7 −44.1 ± 11.1 *p* = 0.45	92.7 92.0 *p* > 0.05	7.1 ± 17.4 6.7 ± 15.9 *p* = 0.88	−28.2 ± 10.7 −29.6 ± 8.4 *p* = 0.30	−9.9 ± 41.7 vs. −11.0 ± 56.9 *p* = 0.68	PIT 2 mg/day was as effective as ATOR 10 mg/day
Yotoke et al., Japan 2008 [41]	12 weeks	177.1 ± 33.4 178.3 ± 33.8	−42.6 ± 12.1 −44.1 ± 11.1 *p* < 0.001	- -	3.2 ± 13.0 ^p1^ 1.7 ± 12.7 ^p2^	−29.7 ± 8.9 −31.1 ± 9.4 *p* < 0.001	−17.3 ± 32.4 ^p3^ −10.7 ± 33.7 ^p4^	PIT 2 mg/day was as effective as ATOR 10 mg/day
Liu et al., Taiwan 2013 [40]	12 weeks	149.6 ± 26.4 151.2 ± 30	−35.0 ± 14.1 −38.4 ± 12.8 *p* < 0.001	50 66 *p* = 0.11	−1.7 ± 11.9 ^p5^ −1.8 ± 11.5 ^p6^	−27.3 ± 10.0 −28.7 ± 9.1 *p* < 0.001	−18.1 ± 32.9 −19.1 ± 26.4 *p* < 0.001	PIT 2 mg/day was as effective as ATOR 10 mg/day
Sasaki et al., Japan 2008 [42]	12 months	163.7± 23.7 161.9 ± 27.6	−33.0 ± 16.1 −40.1 ± 13.5 *p* = 0.002	- -	8.2 ± 17.1 2.9 ± 14.6 *p* = 0.031	- -	−7.1 ± 40.4 −14.6 ± 49.2 *p* = 0.269	PIT 2 mg/day was weaker than ATOR 10 mg/day
Moroi et al., Japan 2020 [44]	12 months	148.4 ± 40.5 149.8 ± 37.0	−36.1 −37.6 *p* > 0.05	-	-	-	-	PIT 2 mg/day was as effective as ATOR 10 mg/day in patients with >1 risk factor for developing ASCVD
Pitavastatin 2–4 mg vs. Atorvastatin 10–20 mg								
Han et al., Korea 2012 [35]	12 weeks	145 ± 40.5 148.8 ± 40.2	−34.6 ± 16.0 −38.1 ± 16.2 *p* < 0.0001	- -	4.6 ± 16.5 3.9 ± 15.2 *p* > 0.05	−26.1 ± 12.4 −28.1 ± 12.5 *p* < 0.0001	−26.7 ± 30.1 −22.3 ± 39.5 *p* = 0.011 **	PIT 2–4 mg/day was as effective as ATOR 10–20 mg/day
Saku et al., Japan 2011 [34]	16 weeks	164 ± 23 162 ± 24	–41 −39 *p* > 0.05	95 94	0 1	- -	−20 −21	PIT 2–4 mg/day was as effective as ATOR 10–20 mg/day
Kurogi et al., Japan 2013 [39]	30 months	122.6 ± 21.2 122.9 ± 23.9	−28.7 −25 *p* = 0.57	- -	11.1 8.1 *p* = 0.013	−14.9 −12.5 *p* = 0.60	−12.8 ^m.e^ −15.8 ^m.e^, *p* = 0.29	PIT 2–4 mg/day was as effective as ATOR 10–20 mg/day
Pitavastatin 2 mg vs. Rosuvastatin 5 mg, 10 mg								
Kong et al., Korea 2017 ^Non-RCT^ [49]	6 months	121.6 ± 20.8 136.3 ± 31.0 143.8 ± 35.6	−41.7 –51.6 −56.0 *p* < 0.05	96.5 ^t2^ 95.0 ^t2^ 95.0 ^t2^	−1.6 –2.1 2.9	–15.2 ± 7.9 –29.9 ± 12.7 –25.7 ± 6.1	–15.9 –8.9 −12.6	PIT 2 mg/day was less effective than both ROS 5 mg/day and ROS 10 mg/day
Pitavastatin 2–4 mg vs. Rosuvastatin 2.5–5 mg								
Saku et al., Japan 2011 [34]	16 weeks	164 ± 23 172 ± 28	−41 −42 *p* > 0.05	95 89	0 4	- -	−20 −19	PIT 2–4 mg/day was as effective as ROS 2.5–5 mg/day
Pitavastatin 2 mg vs. Pravastatin 10 mg, Pravastatin 40 mg, Simvastatin 20 mg								
Saito et al. ^Cl^, Japan 2002 [36]	12 weeks	194.2 (90.5, 297.9) 195.3 (103.8, 286.8)	−37.6 (−39.9, −35.3) −18.4 (−20.7, −16.1) *p* = 0.001	75 (66, 83) ^t3^ 36 (27, 46) ^t3^	8.9 (6.4, 11.4) 9.8 (7.0, 12.6)	−28.0 (−29.6, −26.4) −13.8 (−15.6, −12.0) *p* < 0.001	−23.3 (−32.9, −13.7) * −20.2 (−28.4, −12.0) * *p* = 0.024	PIT 2 mg/day was more effective than PRA 10 mg/day
Kong et al., Korea 2017 ^Non-RCT^ [49]	6 months	121.6 ± 20.8 130.1 ± 18.5	−41.7 –42.4 *p* < 0.05	96.5 91.7	–1.6 2.6	–15.2 ± 7.9 –24.8 ± 10.1	−15.9 −9.4	PIT 2 mg/day was as effective as PRA 40 mg/day
Park et al., Korea 2005 [37]	8 weeks	170.0 ± 28.9 165.7 ± 20.1	−38.2 ± 11.6 −39.4 ± 12.9 *p* = 0.648	93.9 91.3 *p* = 0.709	8.3 ± 13.4 3.6 ± 16.2 *p* = 0.127	−26.9 ± 8.9 −28.5 ± 8.7 *p* = 0.405	−29.8 ± 20.6 −17.4 ± 36.9 *p* = 0.147	PIT 2 mg/day was as effective as SIM 20 mg/day

Abbreviation: HDL-C: high-density lipoprotein cholesterol; LDL-C: low-density lipoprotein cholesterol; TC: total cholesterol; TG: triglyceride; PIT: pitavastatin, ATOR: atorvastatin, ROS: rosuvastatin, PRA: pravastatin, SIM: simvastatin, ASCVD: atherosclerotic cardiovascular disease, SD: standard deviation. ^Non-RCT^: non-randomized controlled trial. *: Patients that had baseline TG ≥ 150 mg/dL. **: p value in PIT group. ^m.e^: median value, ^CI^: 95% confidence interval. ^p1^: *p* = 0.033; ^p2^: *p* = 0.221; ^p3^: *p* < 0.001, ^p4^: *p* = 0.008; ^p5^: *p* = 0.13, ^p6^: *p* = 0.10. ^t1^: patients achieved LDL-C goal based on NCEP-ATP III recommendation. ^t2^: patients achieved LDL-C < 100 mg/dL. ^t3^: patients achieved LDL-C < 140 mg/dL.

**Table 3 healthcare-13-00059-t003:** Mean change in lipid profile when comparing different doses of pitavastatin.

Author	Intervention	Endpoint	LDL-C		HDL-C (%)	TC (%)	TG (%)	Conclusion Focusing on LDL-C Reduction
			Mean Baseline ± SD (mg/dL)	Change (%)				
RCT								
Saito et al., Japan 2011 [43]	PIT 1 mg PIT 2 mg PIT 4 mg	12 weeks	204.8 ± 54.9 198.7 ± 42.3 217.3 ± 70.3	−33.6 ± 11.9 −41.8 ± 10.2 −47.2 ± 12.5 *p* < 0.001	6.8 ± 9.5 ^m^ 5.9 ± 9.5 ^m^ 7.9 ± 10.1 ^m^ *p* > 0.05	−23.0 ± 9.1 −29.1 ± 8.5 −32.5 ± 9.5 *p* < 0.001	−7.7 ± 40.0 −13.6 ± 33.0 −14.7 ± 39.1 *p* < 0.05	PIT was effective in reducing LDL-C. In patients who have not been treated with lipid-lowering medication, utilizing PIT demonstrated a dose-dependent trend
Non-RCT								
Jeong et al., Korea 2020 ^m^ [45]	PIT 1 mg PIT 2 mg PIT 4 mg	8 weeks	131.89 ± 40.18 131.64 ± 45.17 135.29 ± 43.79	–33.34 ± 23.93 –39.13 ± 39.46 –46.27 ± 39.07 *p* < 0.001	4.34 ± 7.42 0.83 ± 11.54 1.10 ± 9.94 *p* < 0.05	–50.01 ± 36.45 –44.73 ± 43.99 –55.47 ± 45.88 *p* < 0.05	–27.93 ± 61.22 –26.16 ± 109.96 –40.54 ± 213.27 *p* < 0.05	PIT was effective in reducing LDL-C. In patients who have not been treated with lipid-lowering medication, utilizing PIT demonstrated a dose-dependent trend
Yamasaki et al., Japan 2014 [48]	PIT 1 mg PIT 4 mg	6 months	143 ± 31 169 ± 34	–26.57 –47.34 *p* < 0.01	5.09 5.08	–20.44 –32.54	–15.65 –18.84	PIT was effective in reducing LDL-C. A high dose was more potent than a low dose.
Koshiyama et al., Japan 2008 [46]	PIT 1 mg (25%) PIT 2 mg (75%)	12 months	153.4 ± 3.3	−30.3	2.6	−15.0	−15.9	PIT was effective in reducing LDL-C

Abbreviation: HDL-C: high-density lipoprotein cholesterol; LDL-C: low-density lipoprotein cholesterol; TC: total cholesterol; TG: triglyceride; PIT: pitavastatin, RCT: randomized controlled trial, SD: standard deviation (±). ^m^: mean change (mg/dL).

**Table 4 healthcare-13-00059-t004:** Data related to adverse drug events, serious adverse events, discontinuation rate, and tolerability of pitavastatin compared to other statin groups.

Safety Sample, Ref	Endpoint	Adverse Drug Reaction				SAE: Rhabdomyolysis (%)	Discontinuation Rate Due to ADR (%)	Tolerability
		Incidence (%)	Severity	Symptoms in Other Organ	Musculoskeletal Symptoms (%)			
Pitavastatin 1 mg vs. Atorvastatin 10 mg								
50 vs. 50 [33]	8 weeks	Low *p* > 0.05	Mild to moderate	Vertigo, nausea, vomiting, headache, stomach ache	Myalgia: 10 vs. 4 Muscle spasm: 2 vs. 0	0	4 vs. 0	Well-tolerated
70 vs. 67 [47]	12 weeks	-	-	-	0 vs. 0	0	-	Well-tolerated
Pitavastatin 2 mg vs. Atorvastatin 10 mg								
136 vs. 132 [38]	8 weeks	10.3 vs. 14.4	-	Headache, dizziness, indigestion, dry mouth, nausea, abdominal pain, diarrhea, constipation	Myalgia: 0.7 vs. 0	0	0.7 vs. 3	Well-tolerated
101 vs. 103 [41]	12 weeks	Low *p* > 0.05	Mild to moderate	-	-	0	-	Well-tolerated
112 vs. 113 [40]	12 weeks	Low *p* > 0.05	-	Skin rash, anxiety, upper airway infection, nasopharyngitis, cough	Myalgia 0.9 vs. 1.8 Back pain: 1.8 vs. 2.7	0	-	Well-tolerated
96 vs. 93 [42]	12 months	9 vs. 14 *p* > 0.05	-	PIT: double or blurred vision, abdominal fullness, nausea, fatigue, headache, pruritus ATOR: appetite loss, diarrhea, abdominal pain, gastritis, stomatitis, angina pectoris, fatigue, headache, hard to sleep, thinking nail	Myalgia: 1 vs. 2 Shoulder stiffness: 0 vs. 1	-	6.2 vs. 6.4	Well-tolerated
312 vs. 310 [44]	12 months	-	-	-	Muscle complaint: 1.3 vs. 3.9	0.3 vs. 0	1.6 vs. 1.0 *p* = 0.725	Well-tolerated
Pitavastatin 2–4 mg vs. Atorvastatin 10–20 mg								
97 vs. 92 [35]	12 weeks	6.2 vs. 6.5 *p* = 1.00	Mild to moderate	Headache, dizziness, anxiety, nausea, vomiting, edema, herpes zoster, urticarial, cough, rhinorrhea	0 vs. 0	-	2 vs. 0	Well-tolerated
99 vs. 99 [34]	16 weeks	17 vs. 18 *p* > 0.05	Mild to moderate	PIT: gastritis, malaise, pruritus ATOR: diarrhea, feeling abnormal, dermatitis, periorbital edema	Myalgia: 1 vs. 0 Muscle spasm: 1 vs. 0	-	5.1 vs. 8.1	Well-tolerated
32 vs. 39 [39]	30 months	4.6 vs. 3.1 *p* = 0.66	-	PIT: skin eruption, fatigue ATOR: fatigue	-	-	-	Well-tolerated
Pitavastatin 2 mg vs. Rosuvastatin 5mg, 10mg								
355 * [49]	6 months	-	-	-	0 vs. 0 vs. 0	-	-	Well-tolerated
Pitavastatin 2–4 mg vs. Rosuvastatin 2.5–5 mg								
99 vs. 99 [34]	16 weeks	17 vs. 11 *p* > 0.05	Mild to moderate	PIT: gastritis, malaise, pruritus ROS: constipation, malaise, pyrexia, eczema	Myalgia: 1 vs. 0 Muscle spasm: 1 vs. 0 Muscle stiffness: 0 vs. 1 Back pain: 0 vs. 2	-	5.1 vs. 4.0	Well-tolerated
Pitavastatin 2 mg vs., Pravastatin 10 mg, Simvastatin 20 mg								
124 vs. 109 [36]	12 weeks	-	Mild to moderate	-	-	0	2.41 vs. 1.83	Well-tolerated
52 vs. 51 [37]	8 weeks	11.6 vs. 23.5 *p* = 0.126	Mild to moderate	PIT: stomach irritation SIM: anxiety, dysgeusia, eye pain in eye and arm, vomit, indigestion, stomach irritation, flatulence, skin rash	Myalgia: 0 vs. 1.96	0	1.9 vs. 7.8	Well-tolerated
Pitavastatin 1 mg vs. Pitavastatin 2 mg vs. Pitavastatin 4 mg								
88 vs. 86 vs. 86 [43]	12 weeks	5.6 vs. 4.7 vs. 5.8	-	Skin rash, constipation, abdominal pain	-	0	-	Well-tolerated
146 vs. 19,292 vs. 8802 [45]	8 weeks	0.43 **	-	Metabolic disorder, general disorder, gastro disorder, nervous system, skin problem, liver disorder,	Myalgia, neck or shoulder pain, muscle pain, somatic pain	0	-	Well-tolerated
34 vs. 29, PIT 1mg vs. PIT 4 mg [48]	6 months	-	-	-	-	0	-	Well-tolerated
178, PIT 1 mg–2 mg [46]	12 months	-	-	-		0	-	Well-tolerated

Abbreviation: ALT: alanine transaminase/alanine aminotransferase; CK: creatine kinase; ADR: adverse event reaction; AST: aspartate aminotransferase, ULN: upper limit of normal, PIT: pitavastatin, ATOR: atorvastatin, ROS: rosuvastatin, PRA: pravastatin, SIM: simvastatin, SAE: serious adverse events. *: Total number of participants. **: An overall value representing pitavastatin 1–2–4 mg.

**Table 5 healthcare-13-00059-t005:** Incidence of patients that had abnormal laboratory test results on liver, muscle, and kidney function between pitavastatin and other comparators.

Safety Sample, Ref	Endpoint	Liver				Muscle		Kidney
		Elevated AST (%)	AST > ×3 ULN (%)	Elevated ALT (%)	ALT > ×3 ULN (%)	Elevated CK (%)	CK > ×10 ULN (%)	Creatinine > ×1.5 ULN (%)
Pitavastatin 1 mg vs. Atorvastatin 10 mg								
50 vs. 50 [33]	8 weeks	-	0	-	4 vs. 0 *p* = 0.495	-	0	-
70 vs. 67 [47]	12 weeks	-	0	-	0	0	0	-
Pitavastatin 2 mg vs. Atorvastatin 10 mg								
136 vs. 132 [38]	8 weeks	-	-	0.7 ^e1^ vs. 1.5 ^e1^	-	0.7 ^e1^ vs. 1.5 ^e1^	0	-
[41]	12 weeks	0 vs. 1.2	-	1.3 vs. 1.2	-	0 vs. 0	-	0
112 vs. 113 [40]	12 weeks	-	0	-	0	-	0	0
95 vs. 97 [34]	16 weeks	4.2 vs. 5.2 *p* > 0.05	-	4.2 vs. 6.2 *p* = 0.043	-	0.0 vs. 1.0 *p* > 0.05	-	-
96 vs. 93 [42]	12 months	-	0	-	2.08 vs. 0 *p* > 0.05	-	0	0
312 vs. 310 [44]	12 months	-	2.2 vs. 2.6 *p* = 0.801	-	2.2 vs. 2.6, *p* = 0.801	0.6 ^e4^ vs. 0 ^e4^ *p* = 0.499	-	4.8 vs. 6.5 *p* = 0.390
Pitavastatin 2–4 mg vs. Atorvastatin 10–20 mg, Rosuvastatin 2.5–5 mg								
97 vs. 92 [35]	12 weeks	-	-	5.2 ^e3^ vs. 5.4 ^e3^ p = 0.86	0.8 *	1.03 vs. 1.08	-	-
95 vs. 97 [34]	16 weeks	4.2 vs. 3.1 *p* > 0.05	-	4.2 vs. 3.1 *p* = 0.043	-	0.0 vs. 1.0 *p* > 0.05	-	-
Pitavastatin 2 mg vs. Pravastatin 10 mg, Simvastatin 20 mg								
124 vs. 109 [36]	12 weeks	-	-	-	1.61 vs. 0.91 *p* < 0.05	4 vs. 2	-	-
52 vs. 51 [37]	8 weeks	0 ^e2^ vs. 2 ^e2^	0	-	-	3.8 ^e1^ vs. 9.8 ^e1^	0	-
Pitavastatin 1 mg vs. Pitavastatin 2 mg vs. Pitavastatin 4 mg								
88 vs. 86 vs. 86 [43]	12 weeks	3.8 *	0	2.3 *	0	5.4 *	0	-
146 vs. 19,292 vs. 8802 [45]	8 weeks	-	0 vs. 0.18 vs. 011	-	0.68 vs. 0.12 vs. 0.10	2.05 ^e3^ vs. 0.10 ^e3^ vs. 011 ^e3^	0	0 vs. 0.72 vs. 0.65

Abbreviation: ALT: alanine transaminase/alanine aminotransferase; CK: creatine kinase; AST: aspartate aminotransferase; ULN: upper limit of normal. *: overall value of pitavastatin 1–2–4 mg. ^e1^: >×2 ULN. ^e2^: <×2 ULN. ^e3^: >×2.5 ULN. ^e4^: >×5 ULN.

## Data Availability

Data were collected from PubMed, Embase, and Cochrane databases.

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
