# Peer review of "Systematic Review on Efficacy, Effectiveness, and Safety of Pitavastatin in Dyslipidemia in Asia"

_healthcare, 2024, doi:10.3390/healthcare13010059_

Round 1
Reviewer 1 Report
Comments and Suggestions for Authors
Please find the attached notes and comments.

Author Response
Thank you for taking the time to contribute to making our manuscript better. Please see the attachment

Reviewer 2 Report
Comments and Suggestions for Authors
This is a review of different aspects of Pitavastatin in Dyslipemia
The review has been well performed and the authors show the results in different ways in order to make evident the results
There are some minor things to consider:
- Introduction: The aspect of safety should also be included in this section
- Methods: I think that more that say Asian countries it should be described as Asian population.
- Discussion:
o One of the main objectives of the review is to asses if there are differences in the efficacy, effectivity and safety of Pitavastatin in Asian population. However, the authors do no address particularly this point in this section.
o In reference to safety, the authors indicate that Pivastatin has the lower incidence of rabdomyolysis, being this affirmation is based in a reference. After checking this reference, even though is consistent, I consider that the calculation based on non-case reports is not very definitive. Maybe, the authors should add more references addressing this point
Author Response

(The authors gave the same response as above.)

Reviewer 3 Report
Comments and Suggestions for Authors
Dear Authors,
Thank you for submitting this manuscript, which I believe addresses an interesting topic in line with the journal’s aims. Below are my comments and suggestions:
INTRODUCTION - RESULTS
- I believe the introduction and results are written appropriately, and I have no further suggestions for revisions.
METHODS:
- In the sentence “The study was conducted in October 2024, based on the Preferred Reporting Items for Systematic Reviews and Meta-Analyses 2020 (PRISMA) review writing guidelines,” there is a significant error. The study was not conducted, but rather the guidelines were followed for reporting the study (reported).
- The research PICO is not defined. Could you clarify why this is missing?
- The methodological quality was not assessed using specific critical appraisal checklists but rather with guidelines for reporting studies. From a methodological standpoint, this is inadequate. It is recommended to use critical appraisal tools, such as those from JBI.
- Only data extraction is mentioned, what about data synthesis?
- Why was the certainty of the evidence not assessed in addition to methodological quality?
DISCUSSION:
- The discussion section presents many results similar to those in the results section. I suggest expanding the discussion by comparing your findings with other international studies and reducing the amount of data presented in this section.
CONCLUSIONS:
- I recommend adding a paragraph on the future implications of your findings.
ADDITIONAL COMMENTS:
- As this is a systematic review, it should have been registered on PROSPERO (Protocol). This has not been done.
- The authors do not provide supplementary files related to the search strategy. This should be provided in full, as it is a systematic review.
- I recommend including grey literature sources to complete the search strategy.
- The PRISMA checklist for reporting this review is not provided.
- The abstract should be restructured, particularly regarding the methodology.
I believe this systematic review has excellent potential, as the different sections are well-written and comprehensive. However, it lacks adequate methodological quality according to the minimum standards for a systematic review. This affects the overall credibility of the work, as rigorous methodology should be the first aspect to evaluate in any study.
Author Response

(The authors gave the same response as above.)

Round 2
Reviewer 1 Report
Comments and Suggestions for Authors
The article is suitable for publication.
Reviewer 3 Report
Comments and Suggestions for Authors
The authors have made significant changes to the methodology. The manuscript may be published